# Influence of pin length on interfacial characteristics and tensile properties of friction-stir-butt-welded AA1060/Q235 clad plates

**Dengfeng Jia**, **Wei Li***, **Tenghui Xia**

Engineering Comprehensive Training Center, Guilin University of Aerospace Technology, Guilin, Guangxi, China

* 2024033@guat.edu.cn

## Abstract

This study explores friction stir butt welding of AA1060/Q235 clad plates to address the chemical industry's need for robust aluminum/steel clad joints. To manage the formation of intermetallic compounds, we employed FSW tools with varying pin lengths, enabling simultaneous welding of the interfacial transition zone and the cladding fusion zone. The pin length significantly influenced joint tensile strength by affecting three key factors: the thickness of the IMC layer, the effective thickness of the steel region, and interfacial microcracking. Optimal tensile strength was achieved when the pin length was adjusted so that its tip precisely reached the Al/steel interface. Although the study is constrained by the number of process parameters, the observed trends illuminate the initial correlations between processing conditions, interfacial characteristics, and mechanical performance. This underscores the necessity of balancing IMC growth with interfacial integrity.

## Introduction

Al/steel clad plates combine the strength of steel with the high specific strength and corrosion resistance of aluminum, making them ideal for lightweight applications in construction, petrochemicals, and shipbuilding. These thick plates, produced through explosive welding, are already employed as transition connectors in ship hulls [1,2]. However, using fusion welding to butt weld two clad plates for structural purposes is uncommon due to the interfacial transition zone (ITZ) often developing defects such as lack of fusion, high residual stress, and excessive intermetallic compounds (IMC) [3,4]. Friction stir welding (FSW), a solid-state process, avoids these issues by avoiding melting and resolidification when joining dissimilar metals [5,6]. Consequently, FSW presents a promising method to expand the use of Al/steel clad plates in engineering applications.

Recent advances in FSW derivatives—arc-enhanced hybrid, magnetic field–assisted, and ultrasonic-assisted processes—have improved material flow and

**Data availability statement:** All relevant data are within the manuscript.

**Funding:** This work was supported by the Middle-aged and Young Teachers' Basic Ability Promotion Project of Guangxi (2023KY0824). URL: http://jyt.gxzf.gov.cn/. The funders had no role in study design, data collection and analysis, decision to publish, or preparation of the manuscript.

**Competing interests:** The authors have declared that no competing interests exist.

suppressed brittle phases. Song [7] reported that ultrasonic-assisted FSW produced a continuous ~1.2 μm IMC layer in Al–steel lap joints and markedly increased joint strength, although the method raised equipment and process complexity. Liu et al. [8] showed that ultrasonic assistance enhances lap-joint formability and performance but did not examine how tool geometry interacts with ultrasonic energy. Li et al. [9] achieved high-quality welds between aluminum alloy and stainless steel tubes using magnetic pulse welding; that technique, however, relies on an axisymmetric electro-magnetic field and is not directly applicable to plate butt joints. Liu et al. [10] applied arc-enhanced hybrid FSW to Al/steel lap joints and observed that joint strength rose and then fell as pin length increased, but they did not quantify the relationship between IMC thickness and strength. Overall, compared with conventional FSW these approaches entail higher equipment costs, greater process complexity, and incompletely understood field-coupling mechanisms. These limitations motivate a focused investigation of conventional FSW parameters—such as pin length—in Al/steel clad plate butt joints.

Achieving adequate tensile strength and robust interfacial bonding in butt-welded Al/steel clad plates for chemical vessels remains challenging. Although research on FSW of these clad plates is still limited, useful insights can be drawn from more extensive studies of dissimilar Al–steel FSW. Ma et al. [11] reviewed the optimiza-tion of FSW parameters (e.g., rotational speed, welding speed, and plunge depth) to obtain an appropriately thick IMC layer and noted that this promotes Al/steel joining. Kaushik et al. [12] reported that larger tool shoulder and pin diameters increase IMC thickness in Al–steel butt joints because they generate higher frictional heat. Chit-turi et al. [13] demonstrated that, in AA5052/SS304 lap joints, tool tilt angle and pin penetration depth influence the IMC layer's thickness and distribution and thus affect joint strength and fracture mode. Together, these studies show that welding parame-ters and tool geometry control joint quality by regulating heat input and, consequently, IMC layer thickness at the interface. Excessive heat input thickens the IMC layer, while insufficient heat leads to poor bonding or lack of fusion; both outcomes mark-edly reduce tensile strength. However, this understanding is based mainly on studies of Al–steel butt or lap joints in monolithic plates, and investigations addressing FSW of Al/steel clad plate butt joints are still scarce.Despite these insights, simultaneous friction stir welding of the interfacial transition zone (ITZ) and cladding fusion zone (CFZ) in Al/steel clad plate butt joints has not been reported, let alone the influence of pin geometry on interfacial microstructure, IMC thickness, and mechanical proper-ties. To investigate this, we designed FSW tools with varying pin lengths to butt-weld AA1060/Q235 clad plates, welding both zones simultaneously. In our experimental setup, pin length and its penetration depth into the steel are coupled, so the "influ-ence of pin length" discussed here inherently reflects this combined effect.

## Experiment

We utilized explosively welded AA1060/Q235 clad plates, each measuring 200×125 mm and consisting of a 4 mm AA1060 cladding and an 8 mm Q235 sub-strate. Tables 1 and 2 detail the chemical compositions and mechanical properties

**Table 1. Chemical composition of AA1060/Q235 clad plate and filler materials (wt.%).**

| Material | Al | Fe | Si | Cu | Mg | Zn | Mn | Ti | V | C | S | P |
|---|---|---|---|---|---|---|---|---|---|---|---|---|
| 1060 | Bal. | 0.35 | 0.25 | 0.05 | 0.03 | 0.05 | 0.03 | 0.03 | – | – | – | – |
| Q235 | – | Bal. | 0.3 | – | – | – | 0.5 | – | – | 0.16 | 0.05 | 0.045 |
| E4303 | – | Bal. | 0.25 | – | – | – | 0.5 | – | – | 0.12 | 0.035 | 0.04 |
| ER50-6 | – | Bal. | 1.0 | 0.3 | – | – | 1.7 | – | – | 0.12 | 0.03 | 0.02 |

**Table 2. Mechanical properties of AA1060/Q235 clad plate and filler materials.**

| Material | Tensile Strength | Yield Strength | Elongation |
|---|---|---|---|
| AA1060/Q235 clad plate | 357MPa | 226MPa | 20% |
| ER50-6 | ≥500MPa | ≥420MPa | ≥22% |
| E4303 | ≥430MPa | ≥330MPa | ≥22% |

of the AA1060/Q235 clad plate and the filler materials. Grooves were precisely machined by CNC milling, following the geometry depicted in Fig 1a. Simulations of the temperature field during butt welding indicated that, on the steel-side first pass, temperatures dropped below 300 °C beyond 5 mm from the weld centerline. To minimize the formation of Fe-Al IMC, we set the groove width on the aluminum cladding side to 10 mm. Before welding, the groove and surrounding area were wire-brushed to remove oxides, followed by degreasing with ethanol and acetone to eliminate oil and moisture. This cleaning sequence is crucial for preventing slag inclusion and porosity.

The welding of the clad plates was performed in three stages. Initially, we employed TIG welding on the steel side due to its concentrated heat, minimal spatter, and narrow heat-affected zone (HAZ). The subsequent second and third passes utilized SMAW (Fig 1b), ensuring the interpass temperature remained below 200°C to safeguard the aluminum cladding. Table 3 details the TIG and SMAW parameters, which were selected based on consumable standards, plate thickness, and practical experience. Post-welding, we milled off the TIG root reinforcement to align the surface flush with the steel substrate (Fig 1c).

A filler plate measuring 200 mm × 10 mm × 4 mm, made from the same alloy as the cladding, was placed in the groove. We used FSW to join it to the aluminum cladding, the steel substrate, and the underlying steel weld. The tool geometry is shown in Fig 2. To avoid confounding effects on heat input and material flow, we fixed the shoulder diameter, pin diameter, and pin shape (tapered threaded). The tool, made of H13 tool steel, was surface-quenched to enhance wear resistance. We estimated the FSW heat input using $Q = \eta K \omega^2 / v$, acknowledging that actual interface temperatures might vary due to differences in thermal conductivity and contact conditions at the Al/steel interface. Since both one-pass and two-pass trials consistently failed to achieve fusion, we implemented three passes to ensure robust Al/steel bonding (Fig 1d).

We performed welding trials using five different pin lengths while keeping all other parameters constant, as shown in Table 4. Consistent with previous studies [14–16], we maintained a fixed tool tilt of 2.5° and a plunge depth of 0.2 mm to ensure proper shoulder contact and forging, which are typical in Al-alloy FSW. We varied the rotational speed (1000, 1200, 1400 rpm) and welding speed (120, 150, 180 mm/min) to optimize surface quality. Based on the observation of a consistent, defect-free surface at the cladding fusion zone, we selected 1200 rpm and 150 mm/min for subsequent welding.

Wire EDM was employed to cut metallographic and tensile specimens from the weld cross-section. For each parameter set, three specimens were tested, with tensile strengths reported as mean ± SD. Interfacial microstructure and morphology were characterized by optical microscopy (OM, Keyence VHX-2000) and scanning electron microscopy (SEM, ZEISS

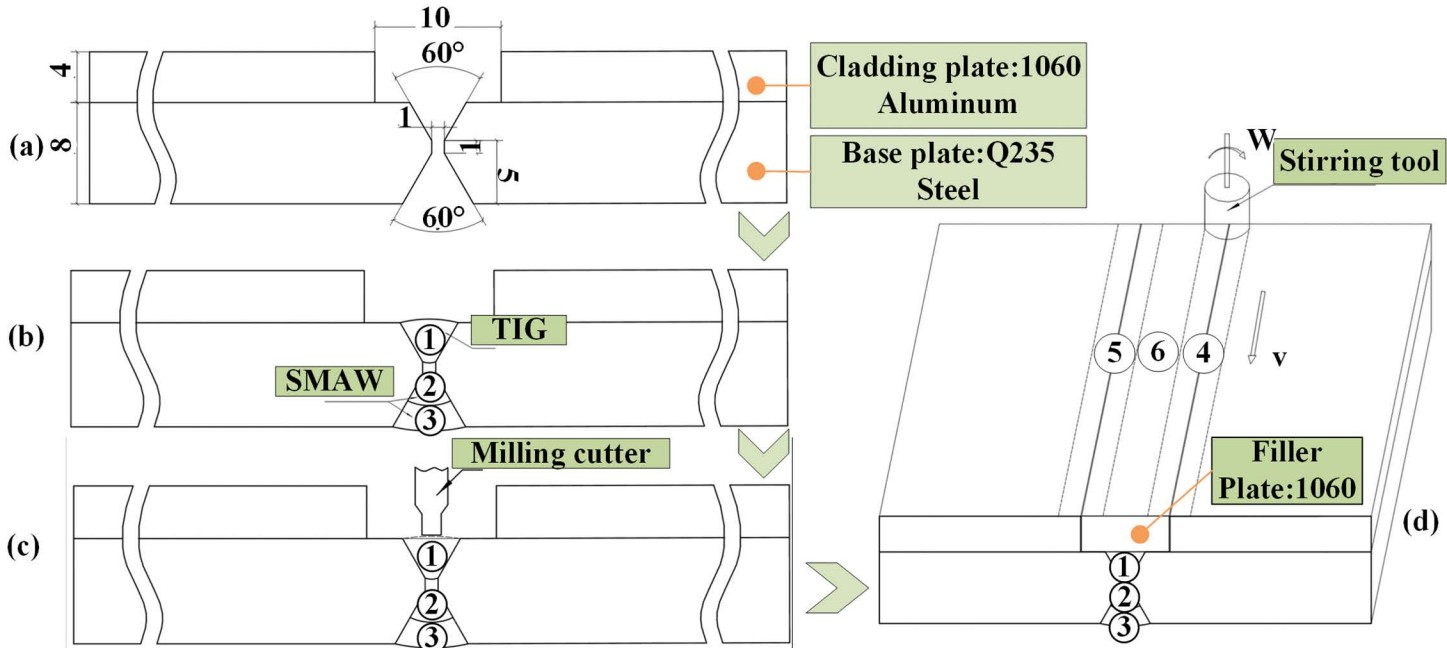

**Fig 1. Welding procedure for the Al/steel clad plates: (a) groove dimensions, (b) welding of the steel substrate, (c) groove milling after steel-side welding, (d) FSW of the aluminum cladding.**

**Table 3. Welding parameters for TIG and SMAW (steel substrate).**

| Welding pass | Welding methods | Current | Voltage | speed | Flow rate of argon in the TIG torch | Wire or Electrode diameter | Welding material type |
|---|---|---|---|---|---|---|---|
| 1 | TIG | 130A | 20V | 7.5 cm/min | 10L/min | 2.0 | ER50-6 |
| 2 | SMAW | 122A | 26V | 14 cm/min | – | 3.2 | E4303 |
| 3 | SMAW | 165A | 28V | 13.3 cm/min | – | 4.0 | E4303 |

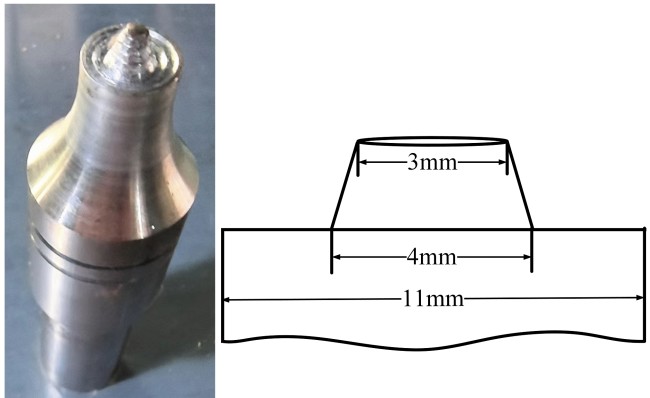

**Fig 2. FSW tool geometry and dimensions.**

**Table 4. FSW welding parameters.**

| Test Plate | Rotating Speed(r/min) | Traversing speed(mm/min) | Press-in amount(mm) | Tilting Angle (°) | Pin length(mm) |
|---|---|---|---|---|---|
| S1 | 1200 | 150 | 0.2 | 2.5 | 3.6 |
| S2 | | | | | 3.8 |
| S3 | | | | | 4.0 |
| S4 | | | | | 4.2 |
| S5 | | | | | 4.4 |

Sigma 300). Elemental profiles across the Al/steel interface were obtained using EDS, while IMC phases were identified by XRD. Tensile tests were conducted according to GB/T 16957−2012 on a universal tester at a displacement rate of 0.5 mm/min. Microhardness across the interface was measured using a DHV-1000Z with loads of 200 g on the Al side and 500 g on the steel side.

## Results and discussion

### Macrostructure of weld appearance

The length of the pin had little effect on weld surface morphology, indicating it is not a primary factor in surface formation. As illustrated in Fig 3, a typical specimen exhibits a smooth weld surface, which suggests sufficient plastic flow, despite the presence of some flash. Further work is needed to quantify surface formation more precisely.

Cross-sectional SEM images of the advancing side (Fig 4) show the impact of pin length on interfacial bonding. With a 3.6 mm pin (Fig 4a), the Al/steel interface exhibits a distinct unbonded region with interfacial separation, indicating poor metallurgical bonding between the filler plate and the steel-side weld. This delamination occurs because the pin tip stops 0.2 mm above the Al/steel interface, generating insufficient frictional heat to plasticize the steel. When the pin length is increased to 3.8 mm (Fig 4b), the pin tip precisely reaches the interface, producing a relatively straight bond line without cutting into the steel. Further increases in pin length from 4.0 mm to 4.4 mm (Figs 4c–4e and 5) lead to pin penetration into the steel rising from 0.2 mm to 0.6 mm. Correspondingly, the interface offset H—defined as the distance between the newly formed FSW interface and the original explosively welded interface—increases from 0.23 mm to 0.77 mm.

The increase in interface offset with longer pins stems from two combined factors: increased frictional heat due to more pin-metal contact and enhanced plasticization of the steel. The inherently lower plastic flowability of steel compared to aluminum intensifies this frictional heating [17], thereby promoting thermal softening. Consequently, H progressively increases with pin length. This offset directly affects the effective thickness of the steel region (ETs), which is defined as the distance from the new Al/steel interface to the steel surface. Fig 5 shows that H and ETs are inversely related: larger H corresponds to smaller ETs. The relationship between heat generation and pin length warrants validation through direct temperature measurements in future work.

### Microstructure of joint interface

Fig 6 shows cross-sectional microstructures of joints made with different pin lengths. At a pin length of 3.6 mm (Fig 6a), the pin stayed completely within the aluminum filler plate, resulting in grain coarsening in the steel-side HAZ due solely to heat conduction. At 3.8 mm (Fig 6b), the pin tip just contacted the interface; mechanical mixing of Al and steel was evident in the bonding zone, while the adjacent steel exhibited partial grain growth and limited recrystallization.

As pin length increased from 4.0 to 4.4 mm, the depth of pin penetration into the steel increased correspondingly from 0.2 to 0.6 mm (Fig 6c–6e). Concurrently, the interface developed graded fiber textures and exhibited incomplete dynamic recrystallization (DRX) in the steel adjacent to the Al/steel interface. Steel fragments and flash also appeared on the

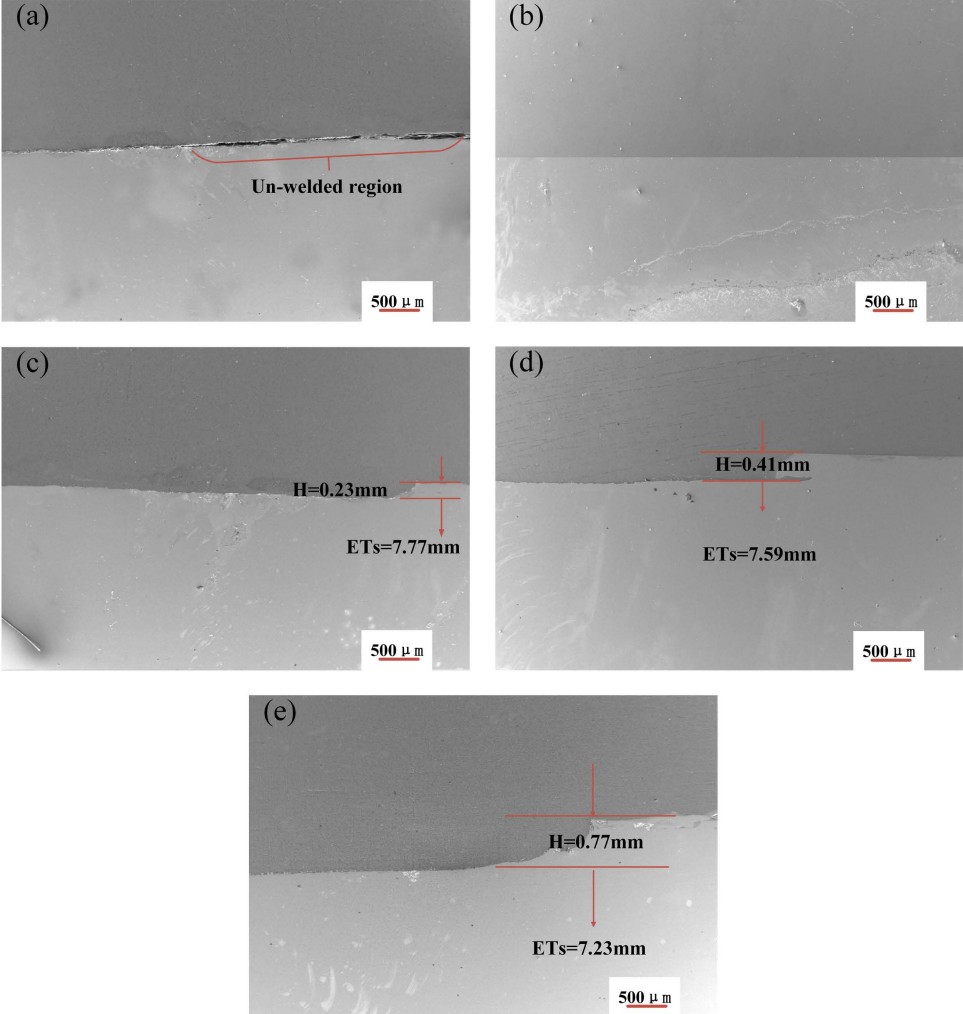

**Fig 3. Surface appearance of a representative FSW weld (pin length = 4.0 mm).**

**Fig 4. Advancing-side weld cross-sections (SEM) for pin lengths of 3.6–4.4 mm: (a) 3.6 mm, (b) 3.8 mm, (c) 4.0 mm, (d) 4.2 mm, (e) 4.4 mm.**

aluminum side due to mechanical cutting by the pin. A magnified view of the steel-side microstructure (Fig 6f) reveals fibrous texture, DRX zones, and regions of incomplete DRX. These microstructural features warrant quantitative EBSD characterization of grain orientation, recrystallization fractions, and texture evolution. A distinct reaction layer with strong phase contrast consistently appeared at the interface, suggesting the formation of Fe-Al IMC. Multiple microcracks were observed within this layer (Fig 6c and 6d), with crack density markedly higher in (d) than in (c). This increase likely results

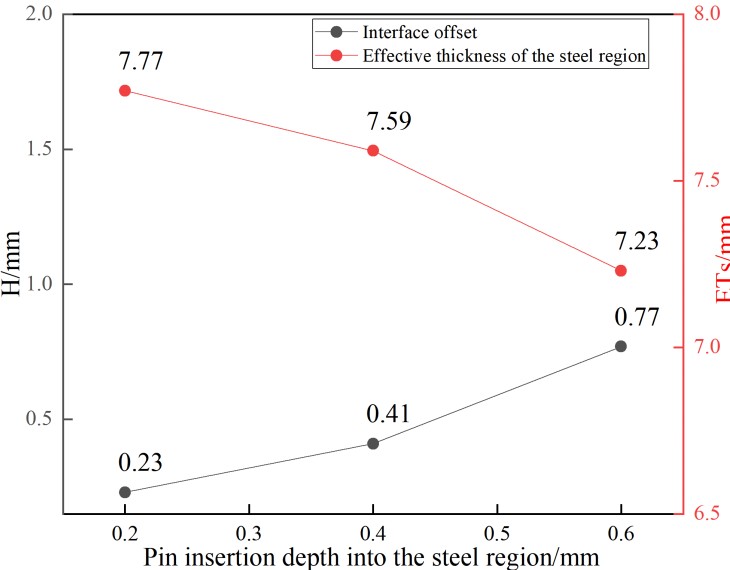

**Fig 5. Effect of pin length on interface offset and effective thickness of the steel region (ETs).**

from greater frictional heat at deeper penetrations, which promotes more extensive Al-Fe interdiffusion and the growth of a brittle IMC layer. Thermal stresses during cooling may then induce cracking within this layer.

To identify the phases present in the reaction layer, we performed EDS point analyses at specific locations near the Al/steel interface in specimen S4, with the locations marked in Fig 7c1. The atomic ratios obtained from these analyses (Table 5) suggest the likely IMC formed. At positions P1 and P4, the Fe:Al ratio is approximately 3:1, suggesting the presence of $Fe_3Al$. At P2, a near 1:1 ratio indicates the formation of AlFe. Meanwhile, at P3, a ratio of about 4:13 aligns with $Fe_4Al_{13}$, a thermodynamically stable phase with minimal Gibbs free energy. XRD analysis (Fig 8) confirmed the formation of these IMCs, corroborating the EDS-based phase identifications. However, EDS point analyses sample only discrete locations, and a single XRD pattern confirms the presence of these phases but does not reveal their spatial distribution or continuity along the entire weld interface.

Fig 7 also quantifies how pin length affects the thickness of the IMC layer. As pin length increases, the Al-Fe interdiffusion distance expands from 1.5μm to 6.1μm, while the IMC layer thickens from 0.38μm to 3.23μm. This trend reflects that longer pins enhance frictional heating, thereby accelerating interdiffusion. That the diffusion zone is consistently wider than the IMC layer is attributable to the limited solubility in Fe-Al solid solutions [18]. The interfacial elemental profiles in Fig 7a2–d2 reveal step-like features whose number and width increase with increasing pin length. This pattern suggests the formation of a substantial and relatively stable IMC layer at the interface.

Longer pins penetrate deeper into the steel, increasing the entrainment of steel fragments into the aluminum region and rendering the IMC layer progressively more irregular (Fig 7a1–7d1). Microcracks appear in the steel region adjacent to the interface, as shown in Fig 7c1 and 7d1, likely resulting from mismatched thermomechanical conditions between frictional heating and mechanical stirring. Both the entrained steel fragments in the aluminum region and the microcracks in the steel region act as severe stress concentrators. While the preceding discussion has addressed the interfacial features governing mechanical performance, the cumulative thermal effects of the three-pass strategy and their microstructural consequences between passes have yet to be investigated. Future work should examine this inter-pass evolution to clarify its role in determining final joint properties.

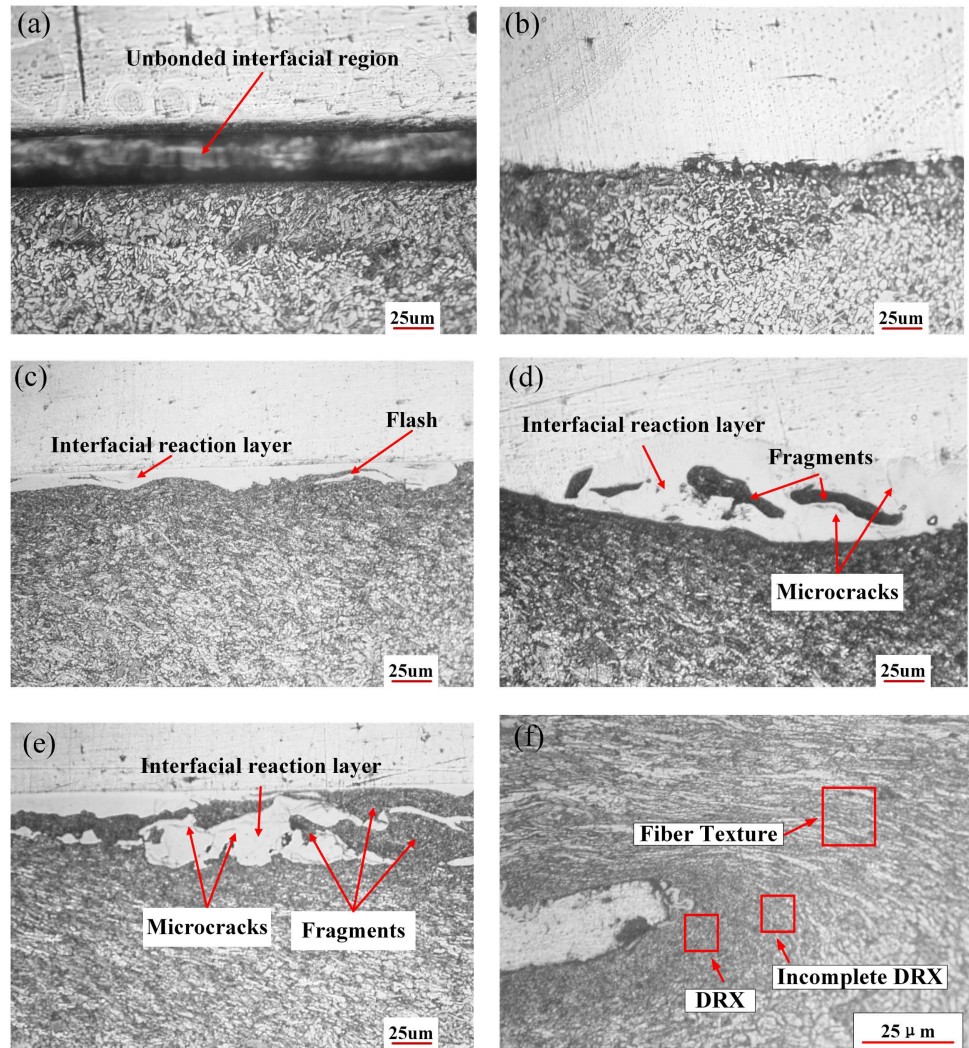

**Fig 6. Microstructure at the Al/steel interface for pin lengths of 3.6–4.4 mm: (a) 3.6 mm, (b) 3.8 mm, (c) 4.0 mm, (d) 4.2 mm, (e) 4.4 mm, with (f) showing a magnified view of the steel-side region adjacent to the interface for the 4.4 mm joint.**

## Mechanical properties

[Fig 9](Fig 9) shows the relationship between joint tensile strength, effective steel thickness, and IMC layer thickness as they change with pin length. Tensile strength varies nonlinearly with pin length. At a pin length of 3.6 mm, the strength reached 367.7 ± 0.37 MPa. Although the steel region maintained its full 8 mm effective thickness, the presence of interfacial gaps and the absence of an IMC layer resulted in minimal mechanical interlocking or metallurgical bonding. Consequently, the steel and aluminum plates bore the load independently during tension. The peak strength of 384.3 ± 0.57 MPa was observed at 3.8 mm. At this length, the steel region's effective thickness remained 8 mm, and a thin (~0.38 μm) continuous IMC layer formed, facilitating both mechanical and metallurgical bonding. This likely contributed to the maximum strength, aligning with previous findings [19].

Beyond this optimal length, tensile strength progressively declined as pin length increased from 4.0 mm to 4.4 mm (365.3 ± 0.12 MPa to 361.2 ± 0.30 MPa). This reduction was linked to two significant microstructural changes: first, the IMC

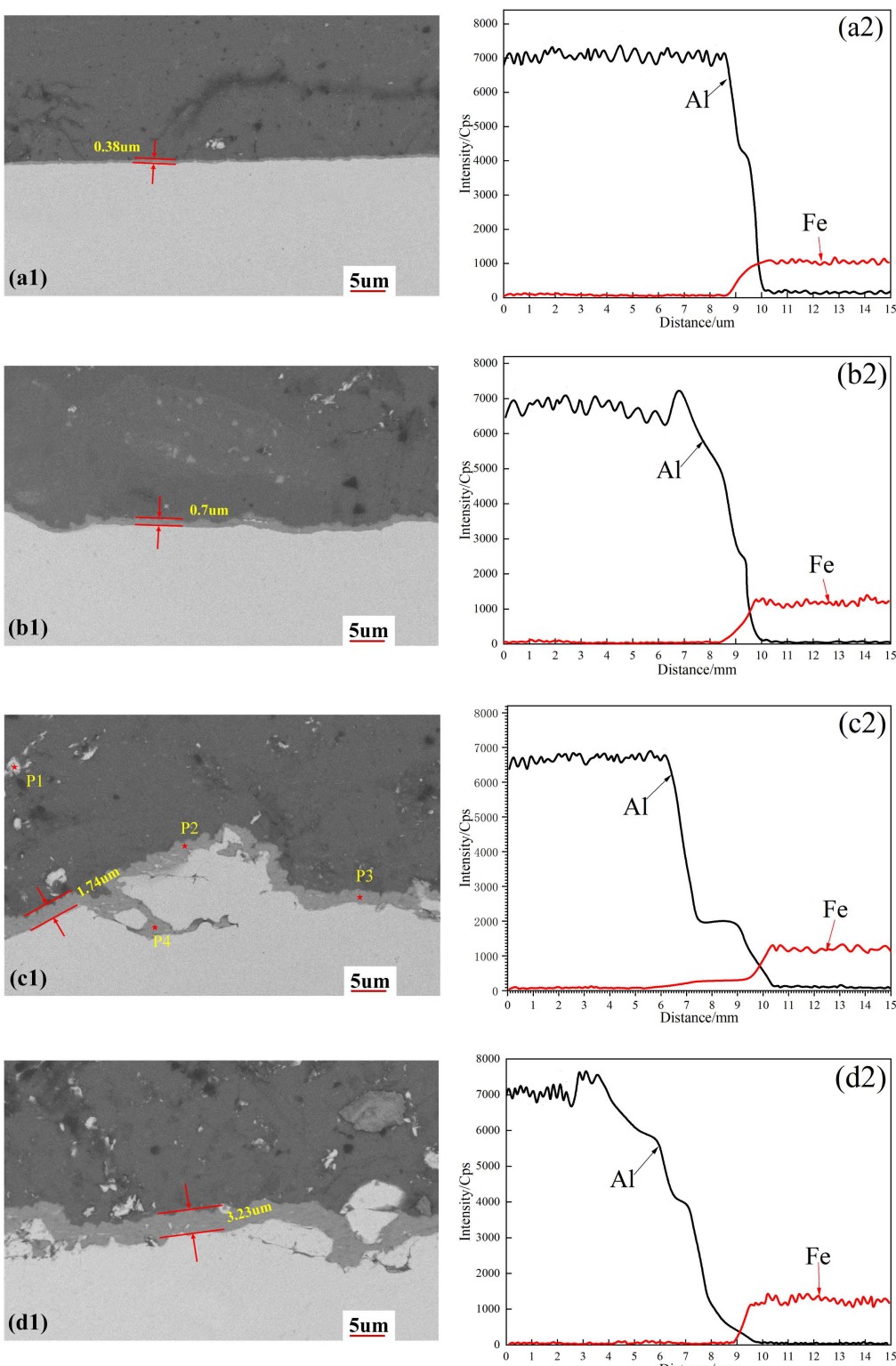

**Fig 7. SEM micrographs (a1–d1) and corresponding EDS line scans (a2–d2) across the Al/steel interface for different pin lengths: (a1, a2) 3.8 mm, (b1, b2) 4.0 mm, (c1, c2) 4.2 mm, (d1, d2) 4.4 mm.**

**Table 5. Chemical composition and possible phases at the points marked in Fig 6c1.**

| Test points | Fe (at%) | Al (at%) | Possible phase |
|---|---|---|---|
| 1 | 86.97 | 13.03 | $Fe_3Al$ |
| 2 | 43.25 | 56.75 | AlFe |
| 3 | 20.37 | 79.63 | $Fe_4Al_{13}$ |
| 4 | 69.27 | 30.73 | $Fe_3Al$ |

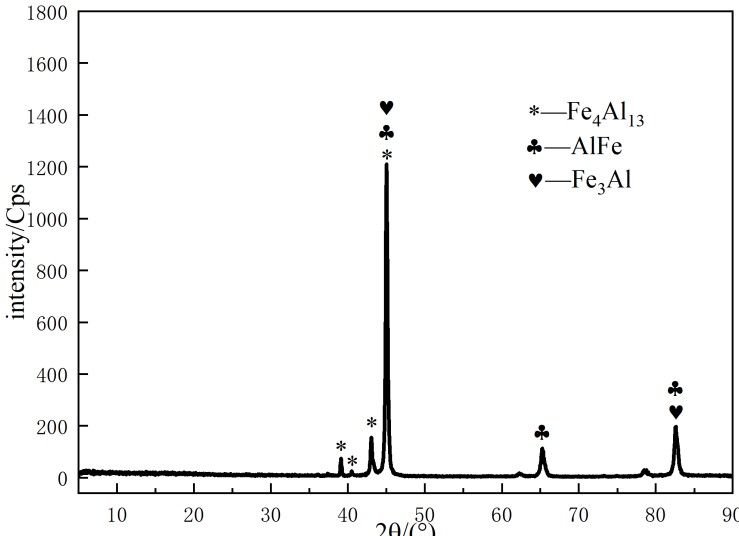

**Fig 8. XRD pattern of the joint cross-section for a pin length of 4.2 mm.**

layer thickened from 0.7 μm to 3.23 μm, with interfacial microcracks and trapped steel fragments indicating increased stress concentration; second, the effective steel thickness decreased from 7.77 mm to 7.23 mm, diminishing its load-bearing capacity. Although these findings suggest a correlation among pin length, interfacial microstructure, and tensile strength, further investigation is needed to establish a clearer causal model. This will include residual stress measurements and a detailed examination of crack propagation mechanisms within the IMC layer.

According to the Chinese standard GB/T 16957-2012, the minimum tensile strength required for clad plates is determined by the following calculation:

$$\sigma_{min} = (\sigma_{b1} \cdot t_1 + \sigma_{b2} \cdot t_2)/(t_1 + t_2)$$

For the AA1060 aluminum cladding layer (thickness $t_1 = 4$ mm), the minimum tensile strength $\sigma_{b1}$ is 110 MPa; for the Q235 steel substrate (thickness $t_2 = 8$ mm), the minimum tensile strength $\sigma_{b2}$ is 375 MPa. Based on this calculation, the minimum tensile strength requirement for the clad plate is determined to be 286.7 MPa. All welded joints in this study exceeded this requirement, with tensile strengths ranging from 361.2 to 384.3 MPa—values even higher than the 357 MPa of the base clad plate. This excellent performance is attributed to the use of high-strength welding consumables for filling the steel-side groove.

Fig 10 presents microhardness profiles measured perpendicular to the interface for joints fabricated with varying pin lengths. Three main features are apparent: (i) minimal hardness variation in the steel and aluminum regions away from

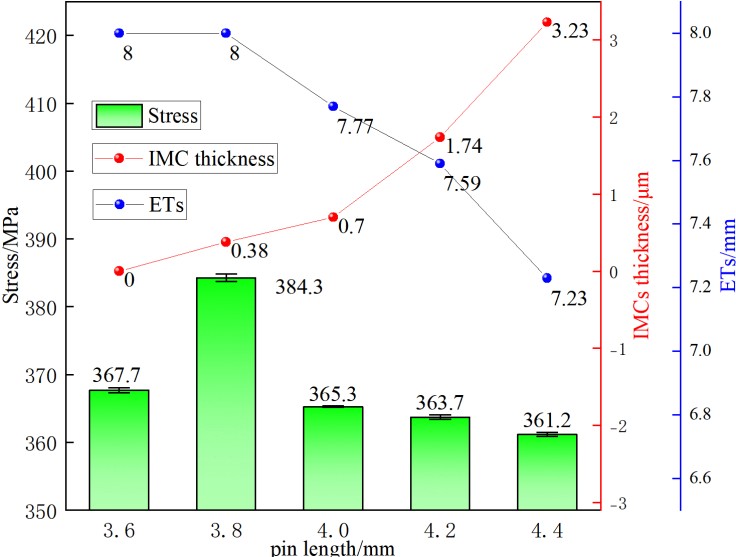

**Fig 9. Tensile strength, effective steel thickness (ETs), and IMC layer thickness as functions of pin length.**

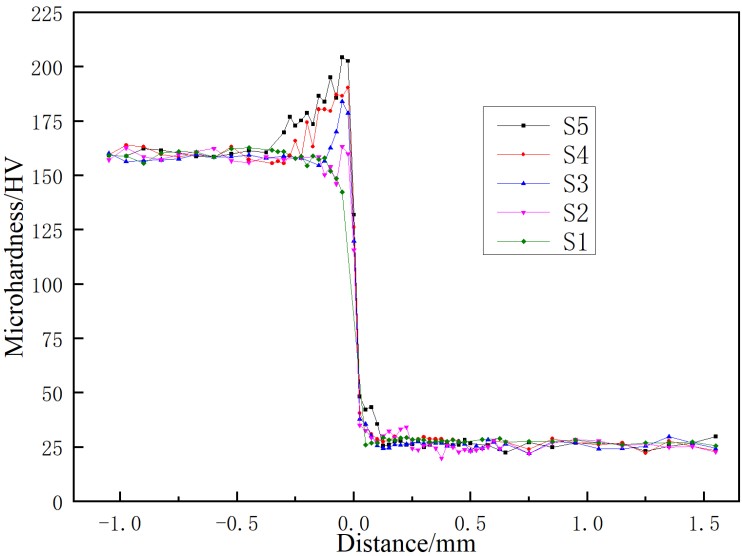

**Fig 10. Microhardness distribution across the joint for various pin lengths.**

the interface; (ii) a noticeable hardness increase in the steel region adjacent to the interface; and (iii) a slight hardness rise in the aluminum region near the interface. These changes are primarily attributed to interfacial microstructural features, specifically grain refinement and IMC formation. Microhardness data for the 3.6 mm pin condition are excluded because this joint lacked effective interfacial bonding.

As the pin length increased, the microhardness at the interface rose from 115.8 HV to 131.95 HV, and the peak hardness on the steel side of the interface increased from 163.3 HV to 204.3 HV. Concurrently, the width of the hardness

gradient zone in the steel, extending from the peak value near the interface to the bulk hardness, expanded from 0.025 mm to 0.275 mm. This broadening is primarily attributed to the thickening of brittle IMC layers and a higher proportion of incompletely dynamically recrystallized grains.

To further understand the microscopic mechanisms behind the observed strength variations, we analyzed the deformation behavior and fracture characteristics of the joints. Fig 11 shows the stress-strain curves. During elastic deformation, the elastic modulus increased slightly with longer pins, aligning with Hooke's law. This increase is likely due to a higher density of heterogeneous particles and dislocations generated at the interface, which impede plastic flow. However, in the subsequent strengthening stage, these particles acted as sites for crack initiation, ultimately reducing both tensile strength and elongation.

To explore the fracture mechanisms at the Al/steel interface, we analyzed tensile fracture surfaces on the steel side adjacent to the interface for various pin lengths (Fig 12). At a pin length of 3.8 mm (Fig 12a), the joint exhibited an elongation of 33.1%, with the fracture surface displaying deep and numerous dimples, indicative of ductile fracture. This suggests that a thin IMC layer improved interfacial bonding. At 4.0 mm (Fig 12b), the fracture mode transitioned to a mixed ductile–brittle type, featuring cleavage river patterns and shallow dimples, with cleavage facets covering about 30% of the surface. Consequently, the elongation decreased to 26.3%, a reduction caused by the excessive thickening of the IMC layer.

Fig 12c and 12d show the fracture surfaces for pin lengths of 4.2 mm and 4.4 mm. At 4.2 mm, distinct cleavage river patterns are evident, whereas at 4.4 mm, the surface shows parallel, sharp cleavage steps. In both cases, cleavage facets make up over 98% of the fracture surface. The elongation values are 24.7% and 22.3%, respectively, which are indicative of brittle fracture. This brittleness is due to excessive IMC thickening and microcracking at the interface. While this study focused on tensile behavior, a comprehensive understanding of joint performance would require additional testing, including evaluations of shear strength and fatigue life.

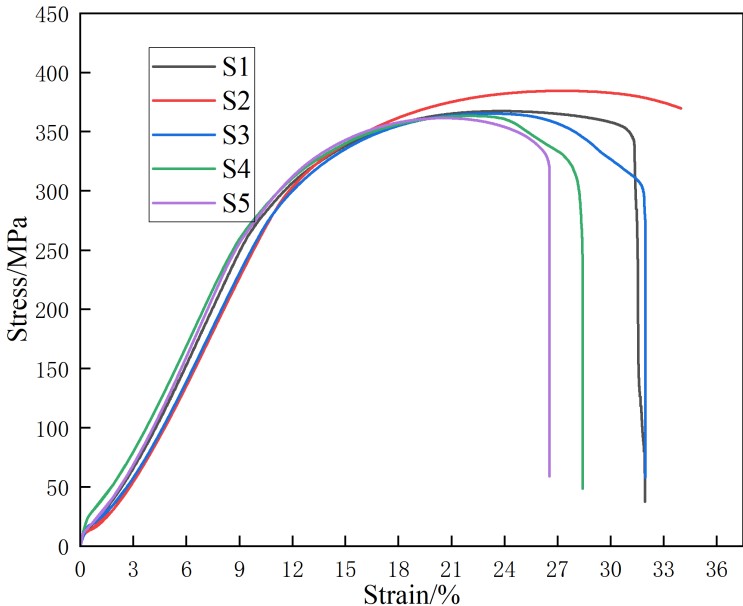

**Fig 11. Stress-strain curves for different pin lengths.**

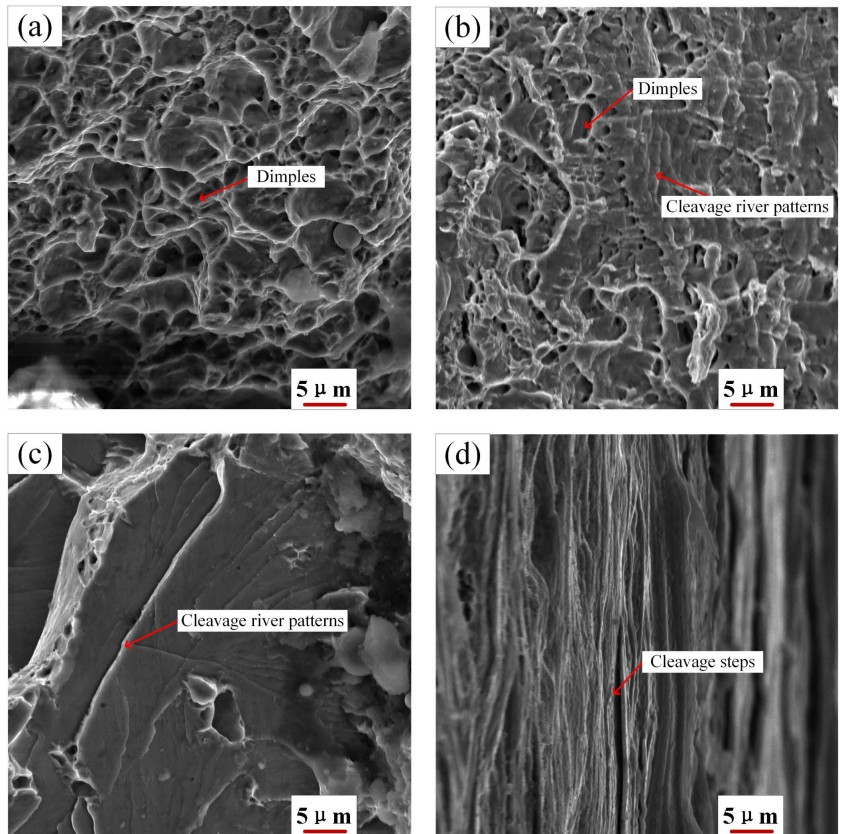

**Fig 12. Fracture surface morphology of the steel region adjacent to the interface for different pin lengths: (a) 3.8 mm, (b) 4.0 mm, (c) 4.2 mm, (d) 4.4 mm.**

## Conclusion

We examined the effect of pin length on the interfacial microstructure and tensile properties of friction stir butt-welded AA1060/Q235 clad plates, while maintaining constant welding speed, rotation speed, and tool geometry. The primary findings are as follows:

(1)  Joint strength is determined by the interplay between interfacial metallurgical bonding and the load-bearing capacity of the steel side. Pin length plays a crucial role in this balance through two coupled mechanisms: the pin's entry into the steel substrate determines the steel's effective thickness; and deeper penetration with longer pins enhances heat input and mechanical stirring, influencing the IMC layer thickness, elemental diffusion distance, and microcrack density at the interface.

(2) Optimal balance is achieved when the pin tip precisely contacts the Al/steel interface. At this length, a continuous submicron IMC layer forms, the steel maintains its full thickness without microcracks, and the joint attains maximum tensile strength. Fracture on the steel side adjacent to the interface exhibits a ductile nature, characterized by a dimpled morphology.

(3) Deviations from this optimal length disrupt the balance. A pin that is too short leaves the steel unplasticized and the interface unbonded; a pin that is too long promotes IMC overgrowth, interfacial microcracking, and reduced steel thickness—all of which lower strength and shift fracture on the steel side to a brittle mode.

Therefore, under the fixed processing conditions applied in this study, positioning the pin tip within ±0.1 mm of the Al/steel interface serves as a practical starting point for optimizing the process. It is important to note that in this study, pin length and penetration depth are inherently coupled. Therefore, the observed optimal pin length is specific to the fixed parameters (e.g., plunge depth, tool geometry) used here and may not directly transfer to other processing conditions or material systems. Future research should focus on decoupling these variables to develop more generalizable process–structure–property models.

## Acknowledgments

The authors express their deep gratitude to the Guangxi Key Laboratory of Special Engineering Equipment and Control Technology for supporting this research.

## Author contributions

**Formal analysis:** Dengfeng Jia.

**Investigation:** Tenghui Xia.

**Validation:** Wei Li.

**Writing – original draft:** Dengfeng Jia.

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
