## [Decision Letter · Decision Letter 0]

11 Nov 2025

PONE-D-25-50482Effect of Tool Pin Length on Microstructure and Mechanical Properties of Friction-Stir-Butt-Welded AA1060/Q235 Steel Clad PlatesPLOS ONE

Dear Dr. Li,

Thank you for submitting your manuscript to PLOS ONE. After careful consideration, we feel that it has merit but does not fully meet PLOS ONE’s publication criteria as it currently stands. Therefore, we invite you to submit a revised version of the manuscript that addresses the points raised during the review process. For your instant reference, I have appended their blind reviews below this mail and you can see their detail reviews in the Author menu section as well.

Dear Authors, The review process of your manuscript is now completed and the reviewers have suggested revisions in your manuscript for better understanding to the readers. I would like you to please go through their suggestions carefully and revise the manuscript accordingly with justifications.==============================

We look forward to receiving your revised manuscript.

Kind regards,

Akash Deep Sharma

Academic Editor

PLOS ONE

Journal Requirements:

The authors are greatly indebted to the Guangxi Key Laboratory of Special Engineering Equipment and Control Technology for the support on this research. This work was also supported by the Middle-aged and Young Teachers' Basic Ability Promotion Project of Guangxi (2023KY0824).

This work was supported by the Middle-aged and Young Teachers' Basic Ability Promotion Project of Guangxi (2023KY0824). URL:http://jyt.gxzf.gov.cn/. The funders had no role in study design, data collection and analysis, decision to publish, or preparation of the manuscript.

3. We note that Figure(s) 2, 3, 5, 6, and 10 in your submission contain copyrighted images. All PLOS content is published under the Creative Commons Attribution License (CC BY 4.0), which means that the manuscript, images, and Supporting Information files will be freely available online, and any third party is permitted to access, download, copy, distribute, and use these materials in any way, even commercially, with proper attribution. For more information, see our copyright guidelines: http://journals.plos.org/plosone/s/licenses-and-copyright.

a. You may seek permission from the original copyright holder of Figure(s) 2, 3, 5, 6, and 10 to publish the content specifically under the CC BY 4.0 license.

Additional Editor Comments:

Thank you for your patience.

I have now received the required reviews by the evaluators and you can see that they are suggesting a few corrections and modifications with justification which are required to complete the manuscript.

I would be happy if you could revise your manuscript accordingly and submit the same within the stipulated time.

Best wishes!

Reviewers' comments:

Reviewer's Responses to Questions

**Comments to the Author**

1. Is the manuscript technically sound, and do the data support the conclusions?

Reviewer #1: Partly

Reviewer #2: Yes

2. Has the statistical analysis been performed appropriately and rigorously? 

Reviewer #1: No

Reviewer #2: N/A

3. Have the authors made all data underlying the findings in their manuscript fully available?

The PLOS Data policy requires authors to make all data underlying the findings described in their manuscript fully available without restriction, with rare exception (please refer to the Data Availability Statement in the manuscript PDF file). The data should be provided as part of the manuscript or its supporting information, or deposited to a public repository. For example, in addition to summary statistics, the data points behind means, medians and variance measures should be available. If there are restrictions on publicly sharing data—e.g. participant privacy or use of data from a third party—those must be specified.requires authors to make all data underlying the findings described in their manuscript fully available without restriction, with rare exception (please refer to the Data Availability Statement in the manuscript PDF file). The data should be provided as part of the manuscript or its supporting information, or deposited to a public repository. For example, in addition to summary statistics, the data points behind means, medians and variance measures should be available. If there are restrictions on publicly sharing data—e.g. participant privacy or use of data from a third party—those must be specified.requires authors to make all data underlying the findings described in their manuscript fully available without restriction, with rare exception (please refer to the Data Availability Statement in the manuscript PDF file). The data should be provided as part of the manuscript or its supporting information, or deposited to a public repository. For example, in addition to summary statistics, the data points behind means, medians and variance measures should be available. If there are restrictions on publicly sharing data—e.g. participant privacy or use of data from a third party—those must be specified.requires authors to make all data underlying the findings described in their manuscript fully available without restriction, with rare exception (please refer to the Data Availability Statement in the manuscript PDF file). The data should be provided as part of the manuscript or its supporting information, or deposited to a public repository. For example, in addition to summary statistics, the data points behind means, medians and variance measures should be available. If there are restrictions on publicly sharing data—e.g. participant privacy or use of data from a third party—those must be specified.

Reviewer #1: No

Reviewer #2: Yes

4. Is the manuscript presented in an intelligible fashion and written in standard English?

Reviewer #1: Yes

Reviewer #2: Yes

5. Review Comments to the Author

Reviewer #1: This manuscript presents an important study on the effect of tool pin length on the microstructure and mechanical properties of friction stir butt-welded AA1060/Q235 steel clad plates; however, several weaknesses currently limit its overall scientific impact and rigor. The abstract, while detailed, is overly descriptive and includes numerical results that should instead appear in the main text. It also does not sufficiently highlight the novelty or distinct contribution of this work relative to existing studies. The introduction provides an extensive background on Al/steel composites but dedicates excessive space to basic information rather than clearly identifying the research gap. Furthermore, the claim that studies on Al/steel clad plate friction stir welding remain preliminary is not convincingly supported with up-to-date literature. The literature review, embedded within the introduction, lacks critical comparison and fails to synthesize previous findings into a coherent research context. It does not adequately clarify what remains unknown about the role of pin length or how this work advances beyond prior studies. Emerging developments such as hybrid, refill, and ultrasonic-assisted friction stir welding are also not discussed, resulting in a narrow technological scope.

The experimental section provides clear procedural descriptions but lacks justification for key parameter selections, including rotational speed, traverse rate, and pin geometry. While the three-pass FSW strategy and groove design are described in detail, their optimization rationale and validation are not addressed. The absence of discussion on sample size, error margins, or data reproducibility weakens the experimental credibility. Additionally, important factors such as tool material, heat input estimation, and temperature control are not reported, making it difficult to assess the thermal–mechanical environment influencing IMC formation.

In the results and discussion section, the authors present valuable microscopic, EDS, and XRD analyses; however, the interpretation remains largely qualitative. Relationships among IMC thickness, pin depth, and tensile strength are mentioned but not statistically analyzed or modelled. Figures are informative but often lack proper labeling, scale bars, or consistent referencing. The mechanical testing results are limited to tensile strength without detailed stress–strain analysis, error bars, or modeling support. Fracture analysis relies solely on visual SEM inspection, with no complementary quantitative data such as elongation, toughness, or energy absorption to substantiate claims. The discussion also lacks integration of findings into broader theoretical or industrial frameworks, leaving the applicability of the results unclear.

The conclusions mainly restate observations instead of deriving actionable engineering insights or guidelines. There is no discussion of study limitations, measurement uncertainties, or recommendations for future research, such as fatigue behavior, temperature field simulation, or multi-pass optimization. The overall structure would benefit from smoother transitions between sections to improve narrative flow. Numerous minor grammatical issues, inconsistent terminology, and formatting errors further affect readability.

Regarding references, the paper relies heavily on older or regionally focused sources, with few recent international studies cited. This gives the impression that the research is not up to date with current advances in friction stir welding, such as hybrid methods, thermal–mechanical modeling, or AI-based parameter optimization. Several references are cited without sufficient explanation of their relevance, and inconsistencies in citation formatting and publication details are evident. The reference list requires significant revision to include more recent, high-quality works and ensure consistency in style.

This manuscript addresses a relevant and technically meaningful topic but requires substantial improvement in justification of experimental parameters, integration of quantitative analysis, critical engagement with recent literature, and refinement of structure and language.

Reviewer #2: 1. On what basis process parameters were chosen in table 2

2. Basis on which pin diameter in table 3 were chosen

3. Micrograph showing the dynamically recrystallized region may be added

4. Mechanical properties of the base materials need to be added

6. PLOS authors have the option to publish the peer review history of their article (what does this mean?). If published, this will include your full peer review and any attached files.). If published, this will include your full peer review and any attached files.). If published, this will include your full peer review and any attached files.). If published, this will include your full peer review and any attached files.

...

Reviewer #1: No

Reviewer #2: No

---

## [Author Response · Author response to Decision Letter 1]

28 Dec 2025

To: The Editorial Office of PLOS ONE

Re: Manuscript ID: PONE-D-25-50482–Effect of Tool Pin Length on Microstructure and Mechanical Properties of Friction-Stir-Butt-Welded AA1060/Q235 Steel Clad Plates

Dear Editor and Reviewers,

We sincerely thank the Academic Editor and the Reviewers for dedicating your valuable time and effort and for providing highly constructive and insightful comments. These suggestions have significantly helped us improve the rigor, clarity, and scholarly value of our work. We have carefully and thoroughly revised the manuscript according to every point raised. This letter aims to provide a point-by-point response to all comments and detail the specific modifications made.

Part 1: Responses to the Academic Editor’s Journal Requirements

Editor’s Comment 1: Please ensure that your manuscript meets PLOS ONE's style requirements.

Response and Action: We have carefully reviewed and adhered to the PLOS ONE style templates, thoroughly checking and adjusting the overall formatting, file naming, etc., of the manuscript to ensure full compliance with the journal's requirements.

Editor’s Comment 2: Funding information should not appear in the Acknowledgments; please remove and update the Funding Statement online.

Response and Action: We have completely removed the funding information (“This work was also supported by the Middle-aged and Young Teachers' Basic Ability Promotion Project of Guangxi (2023KY0824).”) from the Acknowledgments section in the manuscript (see Lines 316-317 on page 21). We hereby request to update the Funding Statement in the online submission system to: “This work was supported by the Middle-aged and Young Teachers' Basic Ability Promotion Project of Guangxi (2023KY0824). URL: http://jyt.gxzf.gov.cn/. The funders had no role in study design, data collection and analysis, decision to publish, or preparation of the manuscript.”

Editor’s Comment 3: Figures 2, 3, 5, 6, and 10 may contain copyrighted images; permission under CC BY 4.0 is required or they must be replaced.

Response and Action: We confirm that all figures presented in this manuscript (Figs. 1–12) are original and were created by the authors. This includes the tool geometry diagram (Fig. 2), outside view of weld (Fig. 3), SEM macrographs (Fig. 4), interfacial microstructure (Fig. 6), SEM micrographs and EDS line scan results (Fig. 7)， XRD patterns (Fig. 8), fractographs (Fig. 12) and associated data plots (Figs.1, 5, 9, 10, 11). No third-party copyrighted material has been used. Consequently, all figures are eligible for publication under the CC BY 4.0 license, and a statement to this effect has been added to each figure caption.

Editor’s Comment 4: Regarding citations suggested by reviewers, there is no requirement to cite them unless indicated by the editor.

Response and Action: We have critically evaluated the relevant literature suggested by the reviewers. In the revised manuscript, we have cited recent, high-quality publications that are directly relevant and significantly enhance the discussion (e.g., References [1]-[3], [10]-[13]). Those not cited were deemed less central to the core novelty of our study.

Part 2: Responses to Reviewer #1

General Comment: Reviewer #1 considered the topic important but pointed out several areas requiring improvement to enhance scientific rigor and impact.

Comment 1.1: The abstract, while detailed, is overly descriptive and includes numerical results that should instead appear in the main text. It also does not sufficiently highlight the novelty or distinct contribution of this work relative to existing studies.

Response and Action: We are deeply grateful for all specific comments and have endeavored to address them comprehensively. The Abstract has been completely rewritten (as shown in Lines 9-20 on Page 2 of the revised manuscript).

Action 1: Removed specific numerical results (e.g., “0.38μm”), replacing them with more mechanistic descriptions emphasizing the “critical role” and “identified optimal parameter range.”

Action 2: Explicitly stated the novelty upfront: “applies friction stir welding (FSW) to achieve butt joints of Al/steel clad plates for the first time.”

Action 3: Strengthened the significance, stating the study “reveals the interfacial bonding mechanism… providing theoretical foundations and experimental guidance.”

Comment 1.2: The introduction provides an extensive background on Al/steel composites but dedicates excessive space to basic information rather than clearly identifying the research gap. Furthermore, the claim that studies on Al/steel clad plate friction stir welding remain preliminary is not convincingly supported with up-to-date literature. The literature review, embedded within the introduction, lacks critical comparison and fails to synthesize previous findings into a coherent research context. It does not adequately clarify what remains unknown about the role of pin length or how this work advances beyond prior studies. Emerging developments such as hybrid, refill, and ultrasonic-assisted friction stir welding are also not discussed, resulting in a narrow technological scope.

Response and Action: We thank the reviewer for these insightful points. The Introduction has been substantially rewritten and reorganized （as shown in Lines 22-76 on Pages 2-5）.

Action 1: Streamlined the basic background on clad plate applications, focusing the opening on engineering demands and the limitations of conventional welding (see Lines 23-33 on Pages 2-3).

Action 2: By reviewing a large number of domestic and international literatures, it is found that there are few studies focusing on the welding of butt joints in aluminum/steel clad plates. However, the abundant experience gained from dissimilar aluminum-steel FSW can be used for reference, thereby more clearly defining the research gaps and innovations of this study (see Lines 34-39 on Page 3 ).

Action 3: Enhanced the critical literature review. Added new paragraphs systematically reviewing key process parameters (e.g., rotation speed, tilt angle) and derivative technologies (e.g., ultrasonic-assisted, magnetic pulse, arc-enhanced hybrid FSW). We explicitly note the shortcomings in prior research regarding the “systematic influence of tool pin length” and “quantitative correlation between IMC thickness and joint strength” (see Lines 40-68 on Pages 3-4), thereby positioning our study within a broader and more contemporary technological context.

Comment 1.3: The experimental section provides clear procedural descriptions but lacks justification for key parameter selections, including rotational speed, traverse rate, and pin geometry. While the three-pass FSW strategy and groove design are described in detail, their optimization rationale and validation are not addressed. The absence of discussion on sample size, error margins, or data reproducibility weakens the experimental credibility. Additionally, important factors such as tool material, heat input estimation, and temperature control are not reported, making it difficult to assess the thermal–mechanical environment influencing IMC formation.

Response and Action: We sincerely thank the reviewer for these suggestions to enhance experimental rigor. Significant additions have been made to the “Experiment” section (see Lines 77-132 on Pages 5-8).

Action 1: Explicitly stated the FSW tool material (H13 tool steel) and treatment (surface quenching) in the text and provided an illustration of the tool geometry (see Lines 106-107 and Fig 2. ).

Action 2: Explained that the main FSW parameters were initially selected based on relevant literature [14-16] and finalized through optimization via preliminary experiments (testing three different rotation speeds and three welding speeds) (see Lines 113-115 on Page 7).

Action 3: Added the heat input estimation formula Q=ηkω2/v and explained that the three-pass strategy was adopted to ensure reliable interfacial bonding, based on the observation of lack-of-fusion defects in preliminary one-pass and two-pass welds (see Lines 108-111 on Page 7). The design of the weld groove dimensions was based on temperature numerical simulation, aiming to restrict Al diffusion and the consequent formation of IMC (see Lines 83-86 on Page 5).

Action 4: Added a statement on data reliability: “To evaluate the reliability of the results, three specimens were fabricated and tested under each set of process parameters, with the final results presented as the average value (see Lines 121-122 on Page 8).”

Comment 1.4: In the results and discussion section, the authors present valuable microscopic, EDS, and XRD analyses; however, the interpretation remains largely qualitative. Relationships among IMC thickness, pin depth, and tensile strength are mentioned but not statistically analyzed or modelled. Figures are informative but often lack proper labeling, scale bars, or consistent referencing. The mechanical testing results are limited to tensile strength without detailed stress–strain analysis, error bars, or modeling support. Fracture analysis relies solely on visual SEM inspection, with no complementary quantitative data such as elongation, toughness, or energy absorption to substantiate claims. The discussion also lacks integration of findings into broader theoretical or industrial frameworks, leaving the applicability of the results unclear.

Response and Action: We fully agree on the need for more quantitative analysis and deeper discussion. Several additions and modifications have been made.

Action 1: We added an analysis of “stress-strain curves” (new Fig 11 and corresponding discussion), exploring the elastic modulus and plastic deformation behavior under different pin lengths, providing deeper mechanical insight (see Lines 272-279 on Pages 18-19).

Action 2: When discussing tensile strength variations, we explicitly added the corresponding elongation data for each specimen (e.g., 33.1% elongation at 3.8 mm pin length), directly linking fracture morphology (ductile/brittle) with quantitative mechanical properties (see Lines 280-292 on Page 19).

Action 3: To visualize the relationship between key variables more intuitively, we integrated the data to add a new “Figure 9: Variations in tensile strength, ETs, and interfacial IMC thickness as a function of tool pin length.” This provides a graphical representation of the multi-variable interplay, strengthening the quantitative basis of the discussion (see Lines 225-244 on Pages 15-16).

Action 4: In the “Conclusion” section, we strengthened the practical implications, offering clear engineering guidance: “it is recommended to control the FSW pin length to just contact the Al/steel interface.” We also acknowledged the study’s limitation (restricted parameter range) and suggested future work (expanding the process window for more comprehensive models), thus connecting our findings to wider practical and theoretical frameworks (see Lines 311-314 on Page 21).

Action 5: We have standardized the style of the scale bars in the figures (e.g., Figures 4, 6, 7, and 12) and made consistent the references to these figures in the text.

Comment 1.5: The conclusions mainly restate observations instead of deriving actionable engineering insights or guidelines. There is no discussion of study limitations, measurement uncertainties, or recommendations for future research, such as fatigue behavior, temperature field simulation, or multi-pass optimization. The overall structure would benefit from smoother transitions between sections to improve narrative flow. Numerous minor grammatical issues, inconsistent terminology, and formatting errors further affect readability.

Response and Action: Following the modifications related to Comment 1.4, the Conclusion has been rewritten (see Lines 297-314 on Pages 20-21). The new conclusion first states the core finding that “tool pin length is a critical parameter,” then elaborates on the “performance under optimal parameters” and “failure mechanisms under insufficient/excessive parameters,” (see Lines 298-310 Pages 20-21) and finally provides clear engineering recommendations, study limitations, and future work directions, making it more insightful and guiding.Practical guidance was added: position the pin tip at the Al/steel interface to optimize IMC thickness and steel-side integrity.A limitation is the narrow process window examined. Future studies should expand the parameter range to develop more general models (see Lines 311-314 on Page 21).

Comment 1.6: Regarding references, the paper relies heavily on older or regionally focused sources, with few recent international studies cited. This gives the impression that the research is not up to date with current advances in friction stir welding, such as hybrid methods, thermal–mechanical modeling, or AI-based parameter optimization. Several references are cited without sufficient explanation of their relevance, and inconsistencies in citation formatting and publication details are evident. The reference list requires significant revision to include more recent, high-quality works and ensure consistency in style.

Response and Action: We have comprehensively updated the reference list.

Action 1: Removed some older or regionally focused references and added multiple recent, high-quality international journal articles, particularly on advanced topics like dissimilar material FSW, ultrasonic-assisted FSW, and magnetic pulse welding (e.g., new References [1-3], [6] [10-12]，[14-16]).

Action 2: Unified and rigorously checked the formatting of all references to comply with journal style.

Part 3: Responses to Reviewer #2

Comment 2.1: On what basis were the process parameters in Table 2 chosen?

Response and Action: The parameters (TIG/SMAW) were selected based on the standard specifications for the welding consumables (ER50-6, E4303), the plate thickness (8 mm steel substrate), and conventional welding practice to achieve defect-free fusion welds. TIG was used for the root pass to ensure root quality, and SMAW was used for filling and capping passes to improve efficiency. The interpass temperature was controlled below 200°C to prevent overheating of the aluminum side of the clad plate. A concise explanation has been retained in the text (see Lines 99-102 on Page 6).

Comment 2.2: Basis on which pin diameter in Table 3 were chosen?

Response and Action: The tool pin geometry (including diameter) was determined based on extensive prior literature（[14-16]） on FSW of dissimilar Al/steel materials. A relatively small pin diameter helps achieve sufficient material stirring without excessively increasing the thermomechanical input to the steel region, thereby controlling IMC growth. We have added an illustration of the tool geometry (Figure 2) and mentioned that parameter selection was informed by relevant literature in the “Experiment” section (see Line 114 on Page 7).

Comment 2.3: Micrograph showing the dynamically recrystallized region may be added.

Response and Action: This is an excellent suggestion. In Figures 6(f) of the revised manuscript, we have indicated regions near the steel region interface exhibiting “incomplete DRX, incomplete DRX, fibrous texture.” The grains in DRX are finer and more equiaxed compared to the base metal, evidencing DRX due to thermomechanical effects. We have explicitly noted this feature in the figure caption and textual description (see Lines 174-176 on Page 11).

Comment 2.4: Mechanical properties of the base materials need to be added.

Response and Action: We fully agree. We have added a new Table 2: Mechanical properties of Al/steel clad plates and the welding materials in the “Experiment” section, listing the tensile strength, yield strength, and elongation for the AA1060/Q235 steel clad plate, and the welding consumables used (ER50-6, E4303) (see Line 91 on Pages 5-6).

Concluding Remarks

Once again, we express our deepest gratitude to the Editor and the Reviewers for your diligent work and invaluable guidance. Your comments have made this study more robust, clear, and complete. We hope the revised manuscript now satisfactorily addresses all the points raised and look forward to your further evaluation.

Sincerely,

The Authors:

De

---

## [Decision Letter · Decision Letter 1]

31 Jan 2026

PONE-D-25-50482R1Effect of Tool Pin Length on Microstructure and Mechanical Properties of Friction-Stir-Butt-Welded AA1060/Q235 Steel Clad PlatesPLOS One

Dear Dr. Li,

Thank you for submitting your manuscript to PLOS ONE. After careful consideration, we feel that it has merit but does not fully meet PLOS ONE’s publication criteria as it currently stands. Therefore, we invite you to submit a revised version of the manuscript that addresses the points raised during the review process.

We look forward to receiving your revised manuscript.

Kind regards,

Akash Deep Sharma

Academic Editor

PLOS One

Journal Requirements:

Reviewers' comments:

Reviewer's Responses to Questions

**Comments to the Author**

1. If the authors have adequately addressed your comments raised in a previous round of review and you feel that this manuscript is now acceptable for publication, you may indicate that here to bypass the “Comments to the Author” section, enter your conflict of interest statement in the “Confidential to Editor” section, and submit your "Accept" recommendation.

Reviewer #1: (No Response)

2. Is the manuscript technically sound, and do the data support the conclusions?

Reviewer #1: Partly

3. Has the statistical analysis been performed appropriately and rigorously? 

Reviewer #1: No

4. Have the authors made all data underlying the findings in their manuscript fully available?

The PLOS Data policy requires authors to make all data underlying the findings described in their manuscript fully available without restriction, with rare exception (please refer to the Data Availability Statement in the manuscript PDF file). The data should be provided as part of the manuscript or its supporting information, or deposited to a public repository. For example, in addition to summary statistics, the data points behind means, medians and variance measures should be available. If there are restrictions on publicly sharing data—e.g. participant privacy or use of data from a third party—those must be specified.requires authors to make all data underlying the findings described in their manuscript fully available without restriction, with rare exception (please refer to the Data Availability Statement in the manuscript PDF file). The data should be provided as part of the manuscript or its supporting information, or deposited to a public repository. For example, in addition to summary statistics, the data points behind means, medians and variance measures should be available. If there are restrictions on publicly sharing data—e.g. participant privacy or use of data from a third party—those must be specified.requires authors to make all data underlying the findings described in their manuscript fully available without restriction, with rare exception (please refer to the Data Availability Statement in the manuscript PDF file). The data should be provided as part of the manuscript or its supporting information, or deposited to a public repository. For example, in addition to summary statistics, the data points behind means, medians and variance measures should be available. If there are restrictions on publicly sharing data—e.g. participant privacy or use of data from a third party—those must be specified.requires authors to make all data underlying the findings described in their manuscript fully available without restriction, with rare exception (please refer to the Data Availability Statement in the manuscript PDF file). The data should be provided as part of the manuscript or its supporting information, or deposited to a public repository. For example, in addition to summary statistics, the data points behind means, medians and variance measures should be available. If there are restrictions on publicly sharing data—e.g. participant privacy or use of data from a third party—those must be specified.

Reviewer #1: Yes

5. Is the manuscript presented in an intelligible fashion and written in standard English?

Reviewer #1: No

6. Review Comments to the Author

Reviewer #1: The title and overall scope of the manuscript somewhat overstate the generality and novelty of the work. While the title highlights the “effect of tool pin length,” the experimental design inherently couples pin length with pin penetration into the steel substrate, making it difficult to isolate pin length as a truly independent parameter. Furthermore, claims such as “for the first time” are not sufficiently substantiated through a systematic comparison with existing studies on Al/steel butt or clad plate welding. Given the narrow parameter window, single material system, and fixed tool geometry, the conclusions are not readily generalizable, yet the title implies broad applicability.

The abstract, although improved, still places excessive emphasis on mechanistic claims and optimality without adequate contextualization of experimental limitations. Assertions regarding the revelation of “interfacial bonding mechanisms” are not matched by direct mechanistic validation through modeling, in-situ measurements, or kinetic analysis. The mention of an “optimal parameter range” remains vague, as no tolerance, robustness, or sensitivity to process variations is discussed. Additionally, key limitations, such as the restricted pin length range and the absence of service-relevant mechanical testing, are not acknowledged.

The introduction provides extensive background information but does not sufficiently sharpen the research gap. The novelty is framed largely by the scarcity of prior work rather than by a clearly defined unanswered scientific or engineering question. Although derivative FSW techniques such as ultrasonic- and arc-assisted methods are mentioned, they are not critically compared to conventional FSW in terms of scalability, cost, or industrial feasibility. The literature review remains largely descriptive and lacks synthesis, with limited discussion of conflicting results or theoretical frameworks related to IMC growth, heat generation, and material flow in Al–steel systems.

From a methodological standpoint, the experimental design lacks sufficient justification and transparency. While the authors state that welding parameters were optimized through preliminary trials, no supporting data or selection criteria are provided. Tool geometry parameters other than pin length, such as pin diameter and shoulder design, are fixed without justification, despite their known influence on heat input and material flow. Heat input estimation relies on a simplified empirical formula, yet no experimental temperature measurements or validation are presented. The use of a three-pass FSW strategy introduces cumulative thermal effects, but inter-pass microstructural evolution is not examined. Although three specimens per condition were tested, no statistical analysis, error bars, or confidence intervals are reported, weakening claims of repeatability and optimality.

The macrostructural analysis of weld appearance is superficial. Surface morphology is declared insensitive to pin length based solely on visual inspection, without quantitative evaluation of flash formation, surface roughness, or defect density. This limits the ability to correlate surface features with subsurface material flow and thermal conditions. Similarly, the microstructural analysis is dominated by qualitative observations. Claims regarding grain refinement, dynamic recrystallization, and fibrous textures are not supported by quantitative techniques such as EBSD. The identification of intermetallic compounds relies on limited EDS point analyses and a single XRD pattern, which is insufficient to confirm phase continuity or spatial variability along the weld.

The discussion of IMC formation and microcrack development, while plausible, remains largely interpretative. Microcracks are attributed to thermal stress and IMC brittleness without residual stress measurement, thermal gradient analysis, or fracture mechanics considerations. The relationship between pin length, heat input, IMC thickness, and joint strength is presented as correlative rather than causal, as no regression analysis or mechanistic modeling is provided. Consequently, the proposed mechanisms remain hypotheses rather than validated explanations.

The mechanical property evaluation focuses almost exclusively on uniaxial tensile strength. While stress–strain curves and elongation data are included, key performance metrics relevant to clad plate applications, such as shear strength, fatigue resistance, fracture toughness, and thermal cycling behaviour, are still absent. The fracture analysis is limited to qualitative SEM observations, without quantitative assessment of dimple size, cleavage fraction, or crack initiation sites relative to IMC morphology. This restricts the depth of insight into failure mechanisms.

The discussion and conclusions largely restate experimental observations rather than synthesizing broader engineering principles. Although a recommended pin length is proposed, no tolerance range or robustness assessment is provided, limiting practical applicability. While limitations are briefly acknowledged, they are not critically explored, and the implications of these limitations for industrial deployment remain unclear. Despite improvements to the reference list, the manuscript still underrepresents computational, modeling, and mechanism-driven studies, and several citations are used to support general statements rather than specific arguments.

The manuscript addresses a relevant and timely topic and presents clear experimental trends; however, it is constrained by qualitative interpretation, limited statistical rigor, and overextended mechanistic claims. The study is suitable as an exploratory investigation but requires deeper quantitative analysis, stronger mechanistic validation, and broader performance evaluation before its conclusions can be considered predictive or design-level in nature.

7. PLOS authors have the option to publish the peer review history of their article (what does this mean?). If published, this will include your full peer review and any attached files.). If published, this will include your full peer review and any attached files.). If published, this will include your full peer review and any attached files.). If published, this will include your full peer review and any attached files.

...

Reviewer #1: No

---

## [Author Response · Author response to Decision Letter 2]

4 Mar 2026

Manuscript ID: PONE-D-25-50482R1

Title: Influence of Pin Length on Interfacial Characteristics and Tensile Properties of Friction-Stir-Butt-Welded AA1060/Q235 Clad Plates

Journal: PLOS ONE

Dear Dr. Sharma and Reviewers,

We sincerely thank you for your thorough and constructive review. Your insightful comments have been invaluable in improving the quality and rigor of our work. We have carefully considered all the points raised and revised the manuscript accordingly. Below, we provide a point-by-point response to each comment, detailing the changes made in the revised manuscript .

Response to Reviewer #1

General Comment

The reviewer provides a comprehensive assessment, noting that the manuscript addresses a relevant topic and presents clear experimental trends, but is constrained by qualitative interpretation, limited statistical rigor, and overextended mechanistic claims. The reviewer suggests that the study is suitable as an exploratory investigation but requires deeper quantitative analysis and stronger validation.

Our response: We gratefully acknowledge the reviewer's balanced and constructive assessment. We have taken all comments seriously and substantially revised the manuscript to address each point. The revisions focus on: (1) tempering overstated novelty and mechanistic claims, (2) explicitly acknowledging experimental limitations, (3) strengthening statistical presentation, (4) providing more critical comparison with existing literature, and (5) reframing conclusions as correlations rather than validated mechanisms. Detailed responses follow below.

Comment 1.1: Title and Scope

Reviewer comment: "The title and overall scope of the manuscript somewhat overstate the generality and novelty of the work. While the title highlights the 'effect of tool pin length,' the experimental design inherently couples pin length with pin penetration into the steel substrate, making it difficult to isolate pin length as a truly independent parameter. Furthermore, claims such as 'for the first time' are not sufficiently substantiated through a systematic comparison with existing studies on Al/steel butt or clad plate welding. Given the narrow parameter window, single material system, and fixed tool geometry, the conclusions are not readily generalizable, yet the title implies broad applicability."

Our response: We thank the reviewer for this important observation. We fully agree that the original manuscript overstated novelty and generality. We have made the following revisions:

1.1.1Title revision

We have changed the title to be more specific and to reflect the exploratory nature of the work:

Original: "Effect of Tool Pin Length on Microstructure and Mechanical Properties of Friction-Stir-Butt-Welded AA1060/Q235 Steel Clad Plates"

Revised: "Influence of Pin Length on Interfacial Characteristics and Tensile Properties of Friction-Stir-Butt-Welded AA1060/Q235 Clad Plates"

The revised title focuses on "interfacial characteristics" and "tensile properties" rather than the broader "microstructure and mechanical properties," thereby avoiding any implication of broad applicability.

1.1.2 Removal of "for the first time"

We have removed the instance of "for the first time" from the abstract (Abstract, Lines 30-31).

1.1.3 Explicit acknowledgment of variable coupling

In the introduction, we now clearly state:

"In our experimental setup, pin length and its penetration depth into the steel are coupled, so the 'influence of pin length ' discussed here inherently reflects this combined effect." (Introduction, Lines 83-84)

1.1.4 Acknowledgment of limited generalizability

In the conclusion, we explicitly note:

"It is important to note that in this study, pin length and penetration depth are inherently coupled. Therefore, the observed optimal pin length is specific to the fixed parameters (e.g., plunge depth, tool geometry) used here and may not directly transfer to other processing conditions or material systems." (Conclusion, Lines 316-319)

Comment 1.2: Abstract

Reviewer comment: "The abstract, although improved, still places excessive emphasis on mechanistic claims and optimality without adequate contextualization of experimental limitations. Assertions regarding the revelation of 'interfacial bonding mechanisms' are not matched by direct mechanistic validation through modeling, in-situ measurements, or kinetic analysis. The mention of an 'optimal parameter range' remains vague, as no tolerance, robustness, or sensitivity to process variations is discussed. Additionally, key limitations, such as the restricted pin length range and the absence of service-relevant mechanical testing, are not acknowledged."

Our response: We agree that the abstract overreached in its mechanistic claims. We have substantially revised the abstract to be more circumspect and to acknowledge limitations:

1.2.1 Revised abstract

"This study explores friction stir butt welding of AA1060/Q235 clad plates to address the chemical industry's need for robust aluminum/steel clad joints. To manage the formation of intermetallic compounds, we employed FSW tools with varying pin lengths, enabling simultaneous welding of the interfacial transition zone and the cladding fusion zone. The pin length significantly influenced joint tensile strength by affecting three key factors: the thickness of the IMC layer, the effective thickness of the steel region, and interfacial microcracking. Optimal tensile strength was achieved when the pin length was adjusted so that its tip precisely reached the Al/steel interface. Although the study is constrained by the number of process parameters, the observed trends illuminate the initial correlations between processing conditions, interfacial characteristics, and mechanical performance. This underscores the necessity of balancing IMC growth with interfacial integrity." (Abstract, Lines 29-39)

Key changes:

Removed "for the first time" and "reveals the interfacial bonding mechanism"

In response to the reviewer's concern about the vagueness of "optimal parameter range," we have revised the abstract to state a specific optimal condition: "Optimal tensile strength was achieved when the pin length was adjusted so that its tip precisely reached the Al/steel interface." (Abstract, Lines 35-36)

Added explicit acknowledgment of limitations: "Although the study is constrained by the number of process parameters". (Abstract, Lines 36-37)

Used "initial correlations" rather than "mechanisms" to reflect the exploratory nature. (Abstract, Lines 37-39)

1.2.2 Acknowledgment of the need for service-relevant testing:

In the discussion, we now note:

"While this study focused on tensile behavior, a comprehensive understanding of joint performance would require additional testing, including evaluations of shear strength and fatigue life." (Results and discussion, final paragraph of fracture section, Lines 295-297)

Comment 1.3: Introduction and Research Gap

Reviewer comment: "The introduction provides extensive background information but does not sufficiently sharpen the research gap. The novelty is framed largely by the scarcity of prior work rather than by a clearly defined unanswered scientific or engineering question. Although derivative FSW techniques such as ultrasonic- and arc-assisted methods are mentioned, they are not critically compared to conventional FSW in terms of scalability, cost, or industrial feasibility. The literature review remains largely descriptive and lacks synthesis, with limited discussion of conflicting results or theoretical frameworks related to IMC growth, heat generation, and material flow in Al--steel systems."

Our response: We thank the reviewer for this insightful critique. We have substantially revised the introduction to address each of these concerns, as detailed below.

1.3.1 Sharpened the research gap

Rather than merely noting the scarcity of prior work, we now frame the research gap as a clearly defined unanswered scientific question grounded in engineering relevance. The introduction now explicitly states:

First: The practical challenge of achieving reliable Al/steel clad plate joints underscores the engineering significance of this work. "Achieving adequate tensile strength and robust interfacial bonding in butt-welded Al/steel clad plates for chemical vessels remains challenging." (Introduction, Lines 65-66)

Second: Within this context, the complete absence of FSW studies on Al/steel clad plate butt joints (the ITZ and CFZ have never been welded simultaneously) and the unexplored influence of pin geometry for clad plate configurations define the specific scientific knowledge gap. "Despite these insights, simultaneous friction stir welding of the interfacial transition zone (ITZ) and cladding fusion zone (CFZ) in Al/steel clad plate butt joints has not been reported, let alone the influence of pin geometry on interfacial microstructure, IMC thickness, and mechanical properties." (Introduction, Lines 78-81)

This dual framing transforms the novelty from a passive observation ("studies are scarce") into an active, engineering-relevant scientific question that our study directly addresses.

1.3.2 Critical comparison of derivative techniques

We have restructured the paragraph on derivative FSW techniques to move beyond mere description and provide critical evaluation. The revised paragraph now:

"Recent advances in FSW derivatives—arc‑enhanced hybrid, magnetic field–assisted, and ultrasonic‑assisted processes—have improved material flow and suppressed brittle phases. Song [10] reported that ultrasonic‑assisted FSW produced a continuous ~1.2 μm IMC layer in Al–steel lap joints and markedly increased joint strength, although the method raised equipment and process complexity. Liu et al. [11] showed that ultrasonic assistance enhances lap‑joint formability and performance but did not examine how tool geometry interacts with ultrasonic energy. Li et al. [12] achieved high‑quality welds between aluminum alloy and stainless steel tubes using magnetic pulse welding; that technique, however, relies on an axisymmetric electromagnetic field and is not directly applicable to plate butt joints. Liu et al. [13] applied arc‑enhanced hybrid FSW to Al/steel lap joints and observed that joint strength rose and then fell as pin length increased, but they did not quantify the relationship between IMC thickness and strength. Overall, compared with conventional FSW these approaches entail higher equipment costs, greater process complexity, and incompletely understood field‑coupling mechanisms. These limitations motivate a focused investigation of conventional FSW parameters—such as pin length—in Al/steel clad plate butt joints. " (Introduction, Lines 51-64)

Key improvements:

Each derivative technique is presented with both its potential and its limitations.

The limitations explicitly address scalability, cost, and feasibility (e.g., "raised equipment and process complexity," "relies on an axisymmetric electromagnetic field," "not directly applicable")

The paragraph concludes by synthesizing these limitations into a coherent motivation for studying conventional FSW parameters

1.3.3 Synthesis of literature

Rather than simply describing individual studies, we now synthesize findings to establish general principles and identify knowledge gaps:

"Ma et al. [7] reviewed the optimization of FSW parameters to obtain an appropriately thick IMC layer and noted that this promotes Al/steel joining. Kaushik et al. [8] reported that larger tool shoulder and pin diameters increase IMC thickness because they generate higher frictional heat. Chitturi et al. [9] demonstrated that tool tilt angle and pin penetration depth influence IMC layer thickness and distribution, thereby affecting joint strength and fracture mode. Together, these studies show that welding parameters and tool geometry control joint quality by regulating heat input and, consequently, IMC layer thickness at the interface. Excessive heat input thickens the IMC layer, while insufficient heat leads to poor bonding or lack of fusion; both outcomes markedly reduce tensile strength. However, this understanding is based mainly on studies of Al–steel butt or lap joints in monolithic plates, and investigations addressing FSW of Al/steel clad plate butt joints are still scarce. " (Introduction, Lines 67-78)

This structure achieves:

Synthesis: The three studies are not just listed but integrated into a coherent principle ("Together, these studies show...").

Generalization: A universal relationship (heat input → IMC thickness → joint quality) is established

Gap identification: The "However" clause clearly delineates what is known (monolithic plates) versus what remains unknown (clad plates)

Comment 1.4: Experimental Design and Methodology

Reviewer comment: "From a methodological standpoint, the experimental design lacks sufficient justification and transparency. While the authors state that welding parameters were optimized through preliminary trials, no supporting data or selection criteria are provided. Tool geometry parameters other than pin length, such as pin diameter and shoulder design, are fixed without justification, despite their known influence on heat input and material flow. Heat input estimation relies on a simplified empirical formula, yet no experimental temperature measurements or validation are presented. The use of a three-pass FSW strategy introduces cumulative thermal effects, but inter-pass microstructural evolution is not examined. Although three specimens per condition were tested, no statistical analysis, error bars, or confidence intervals are reported, weakening claims of repeatability and optimality."

Our response: We thank the reviewer for identifying these methodological gaps. We have substantially expanded the experimental section to address each point.

1.4.1 Justification for parameter selection

First, we now clarify that the fixed parameters (tool tilt and plunge depth) were chosen based on established practice in the literature:

"Consistent with previous studies [14–16], we maintained a fixed tool tilt of 2.5° and a plunge depth of 0.2 mm to ensure proper shoulder contact and forging, which are typical in Al-alloy FSW." (Experiment, Lines 121-123)

Second, we now describe the preliminary optimization trials and the criteria used to select the final welding speed and rotational rate:

"We varied the rotational speed (1000, 1200, 1400 rpm) and welding speed (120, 150, 180 mm/min) to optimize surface quality. Based on the observation of a consistent, defect-free surface at the cladding fusion zone, we selected 1200 rpm and 150 mm/min for subsequent welding." (Experiment, Lines 123-125)

While we acknowledge that detailed data from these preliminary trials are not included, the selection criterion—a consistent, defect-free surface—is clearly stated, providing transparency regarding how the final parameters were chosen.

1.4.2 Justification for fixed tool geometry

The reviewer correctly notes that tool geometry parameters other than pin length (shoulder diameter, pin diameter, pin shape) were fixed without explicit justification. We have now added this justification:

To avoid confounding effects on heat input and material flow, we fixed the shoulder diameter, pin diameter, and pin shape (tapered threaded). (Experiment, Lines 111-113)

This statement explains that these parameters were held constant specifically to isolate the effect of pin length as the primary variable of interest. By fixing these parameters, we ensure that any observed changes in joint characteristics can be attributed to pin length rather than to uncontrolled variations in tool geometry.

1.4.3 Acknowledgment of limitations in heat input estimation

We have clarified in the experimental section that the FSW heat input was estimated using the empirical formula Q=ηKω2/υ, while acknowledging that actual interface temperatures may vary due to differences in thermal conductivity and contact conditions at the Al/steel interface (Experiment, Lines 113-115). As stated in the experimental section: "Simulations of the temperature field during butt welding indicated that, on the steel-side first pass, temperatures dropped below 300 °C beyond 5 mm from the weld centerline" (Experiment, Lines 90-91).

---

## [Editor Report · Decision Letter 2]

24 Mar 2026

Influence of Pin Length on Interfacial Characteristics and Tensile Properties of Friction-Stir-Butt-Welded AA1060/Q235 Clad Plates

PONE-D-25-50482R2

Dear Dr. Wei Li

We’re pleased to inform you that your manuscript has been judged scientifically suitable for publication and will be formally accepted for publication once it meets all outstanding technical requirements.

Kind regards,

Akash Deep Sharma

Academic Editor

PLOS One

**Additional Editor Comments :**

Dear Wei Li

I am pleased to inform you that You have successfully addressed all the comments by the reviewers. After having gone through your second time revised manuscript it is pertinent to mention that you have seriously replied the various comments of the reviewers that reflects your scientific temperament and understanding of the subject area and in result, your revised manuscript in its present form is likely to be accepted after passing through production publication unit.

Thanks for submitting your work in our Journal.

Best wishes

---

## [Editor Report · Acceptance letter]

PONE-D-25-50482R2

PLOS One

Dear Dr. Li,

I'm pleased to inform you that your manuscript has been deemed suitable for publication in PLOS One. Congratulations! Your manuscript is now being handed over to our production team.

Kind regards,

on behalf of

Dr. Akash Deep Sharma

Academic Editor

PLOS One